# NEXUS: NEIGHBORHOOD-ENHANCED CORRESPONDENCE OPTIMIZATION STRATEGY FOR SHAPE CORRESPONDENCES

## ABSTRACT

Shape correspondence is a cornerstone of computer graphics, enabling applications such as shape registration, deformation transfer, and animation. We introduce NEXUS (Neighborhood-Enhanced Correspondence Optimization Strategy), a novel framework that integrates local and global optimization to address the shape correspondence problem effectively. Our primary contribution is the Local Neighborhood Consistency (LNC) metric, a computationally efficient and robust measure for assessing correspondence quality using mesh connectivity rather than geodesic distances. Unlike prior metrics like Local Map Distortion (LMD), LNC is faster to compute (linear in number of edges in mesh adjacency), and is more resilient to non-isometric deformations. We couple LNC with a seeded graph matching approach to refine correspondences, achieving superior accuracy and speed compared to existing methods. Experimental results demonstrate NEXUS's effectiveness across diverse datasets, including near-isometric, non-isometric, and topologically noisy shapes. We also address implementation errors in prior LMD-based methods and highlight NEXUS's limitations, such as sensitivity to significant mesh connectivity discrepancies. Our work simplifies and accelerates shape correspondence pipelines while maintaining or improving accuracy.

## 1 INTRODUCTION

Establishing correspondences between 3D shapes is fundamental to computer vision and graphics applications, including shape retrieval, comparison, recognition, registration, motion, style, and deformation transfer (Hartwig et al., 2023; Eisenberger et al., 2020a; Ren et al., 2018; Xu & King, 2001; Eisenberger et al., 2023; Lähner et al., 2016; Sahillioğlu & Yemez, 2012; Aflalo et al., 2016). However, shape correspondence remains challenging due to its formulation as a Quadratic Assignment Problem (QAP) or Linear Assignment Problem (LAP), both NP-hard (Hartwig et al., 2023; Bastian et al., 2023; Amberg et al., 2007). The complexity increases with deformation types, such as isometric (preserving geometric properties) (Lipman & Funkhouser, 2009; Xiang et al., 2021; Pai et al., 2021; Melzi et al., 2019) or non-isometric (altering angles, distances, or connectivity) (Kim et al., 2011; Bastian et al., 2023; Hartwig et al., 2023; Eisenberger et al., 2020a).

This paper introduces NEXUS (Neighborhood-Enhanced Correspondence Optimization Strategy), a novel framework that advances shape correspondence through a joint local and global optimization approach. The name NEXUS reflects the algorithm's core strength: it connects local neighborhood consistency with global graph-based refinement to achieve robust and efficient correspondence matching. Our primary contributions are:

- **Local Neighborhood Consistency (LNC) Metric**: We propose LNC, a new metric to evaluate correspondence quality using mesh connectivity. Unlike the Local Map Distortion (LMD) (Xiang et al., 2021), which relies on computationally expensive geodesic distances, LNC is computed in $O(nv)$ time, where $n$ is the number of vertices, and $v$ is the number of nonzero elements in the degree matrix, making it significantly faster. LNC is also more robust to non-isometric deformations, as it does not depend on geometric properties like distances or areas. We also identify and correct errors in prior LMD implementations (Xiang et al., 2021; Kamhoua et al., 2022; Kamhoua & Qu, 2024), ensuring accurate baseline

comparisons. These corrections reveal performance degradation in prior methods on certain datasets, underscoring the need for robust metrics like LNC.

- **NEXUS Framework**: We Follow HOPE (Kamhoua & Qu, 2024) and integrate LNC with a seeded graph matching approach (YU et al., 2021) to form NEXUS, which refines correspondences iteratively. NEXUS outperforms state-of-the-art methods in accuracy and speed across diverse datasets, including those with topological noise and non-isometric deformations.[1]

- **Comprehensive Evaluation**: We conduct extensive experiments on datasets like TOPKIDS, SCAPE, TOSCA, and SHREC16, demonstrating NEXUS's generalization and efficiency. We also analyze its limitations, particularly its sensitivity to significant mesh connectivity discrepancies.

These contributions address key limitations in prior work, such as computational inefficiency and sensitivity to non-isometric deformations, while providing a simple, fast, and effective solution for shape correspondence.

## 2 RELATED WORK

Shape matching has been extensively studied, with approaches ranging from traditional optimization to deep learning methods. Below, we review optimization-based techniques and integrate deep learning approaches for a comprehensive overview. For further details, see surveys by Sahillioğlu (2020); Van Kaick et al. (2011); Tam et al. (2013); Biasotti et al. (2016).

**Optimization-Based Correspondence Initialization.** Initial correspondences are often established using user-defined landmarks (Melzi et al., 2019; Shamai & Kimmel, 2017) or by aligning descriptors, either pair-wise (via QAP) using mesh connectivity spectra (Umeyama, 1988; Fan et al., 2020; Feizi et al., 2020; Finke et al., 1987; Kazemi et al., 2016; Dym et al., 2017; Sandryhaila & Moura, 2013), geodesic distances (Xiang et al., 2020; Aflalo et al., 2016), or mass matrices (Xiang et al., 2020), or point-wise (via LAP) using descriptors like Heat Kernel Signatures (HKS) (Bronstein & Kokkinos, 2010), Wave Kernel Signatures (WKS) (Aubry et al., 2011), Geodesic Distance Descriptors (GDD) (Shamai & Kimmel, 2017), Global Point Signature (GPS) (Ovsjanikov et al., 2008), or SHOT (Tombari et al., 2010). These initializations are refined for accuracy and smoothness.

**Spectral-Based Correspondence Refinement.** Spectral methods embed correspondences in the Laplace Beltrami Operator (LBO) basis (Ovsjanikov et al., 2012), relaxing QAP to LAP. Techniques like ZoomOut (Melzi et al., 2019) increase basis resolution iteratively, while others enforce cycle consistency (Huang et al., 2020; Pai et al., 2021) or geometric constraints (Rodolà et al., 2017; Eisenberger et al., 2020a; Ren et al., 2018; Cao et al., 2023a; Sharp et al., 2022a). DIR (Xiang et al., 2021) and GEM (Kamhoua et al., 2022) use LMD to select well-matched points, but LMD's reliance on geodesic distances limits its efficiency and robustness. We propose LNC to address these issues.

**Graph-Based Correspondence Refinement.** Graph-based methods refine correspondences by maximizing neighborhood agreement (Kazemi et al., 2015; YU et al., 2021; Lubars & Srikant, 2018; Kuhn, 2012). HOPE (Kamhoua & Qu, 2024) uses LMD to identify poorly matched points and refines them via seeded graph matching, avoiding functional map limitations. We enhance this approach with LNC, improving efficiency and robustness.

**Deep Learning Approaches.** Deep learning has advanced shape matching by learning complex features. **Supervised methods** use labeled data for high accuracy but require costly annotations. *Deep Functional Maps* (FMNet) (Litany et al., 2017b) optimizes spectral descriptors for functional map alignment, offering robust correspondences but needing extensive labeled data. *3D-CODED* (Groueix et al., 2018) predicts deformation parameters, excelling in non-rigid matching but requiring templates and labeled data. *DGCNN* (Wang et al., 2019) uses dynamic graph convolutions, capturing geometric features but struggling with sparse point clouds. **Unsupervised methods** leverage intrinsic shape properties. *Unsupervised Learning of Robust Spectral Shape Matching* (Marin et al., 2023)

---

[1]code attached to supplementary material.

aligns spectral descriptors using cycle-consistency, scaling well but assuming non-noisy spectra which is often not the case in the presence of topological noise and other mesh edits. *Deep Shells* (Eisenberger et al., 2020b) optimizes shell-based energy, robust to topological changes but less effective for volumetric shapes. *DiffusionNet* (Sharp et al., 2022b) uses diffusion processes, offering fast inference but oversmoothing fine details. **Semi-supervised methods** balance accuracy and scalability. *Semi-Supervised Shape Matching with Pseudo-Labels* (Chen et al., 2022) uses self-supervised pre-training and pseudo-labels, reducing annotation needs but relying on pseudo-label quality. *Graph-Based Semi-Supervised Shape Correspondence* (Xu et al., 2023) propagates labels via graph neural networks, efficient but sensitive to graph quality. Moreover, **all these Deep learning baselines** often need training and retraining when there is a domain shift in the test datasets as a model trained on nearly isometric shapes for example will not perform well on partial shapes (see Cao et al. (2023b)). NEXUS addresses these limitations by needing no training nor retraining.

## 3 PROBLEM DEFINITION AND PRELIMINARIES

A 3D shape $\mathcal{S}$ with $n$ vertices and $f$ faces is represented by vertex locations $\mathbf{X} \in \mathbb{R}^{n \times 3}$ and a mesh via a face matrix $\mathbf{F} \in \mathbb{R}^{f \times 3}$ or adjacency matrix $\mathbf{A} \in \mathbb{R}^{n \times n}$. The Uniform Laplacian is $\mathbf{L_U} = \mathbf{D} - \mathbf{A}$, where $\mathbf{D} = \sum_i [\mathbf{A}]_{i,j}$. The Cotangent Laplacian is $\mathbf{L} = \mathbf{M}^{-1} \mathbf{W}$, with $\mathbf{M}$ as the diagonal matrix of lumped area elements and $\mathbf{W}$ as the cotangent weight matrix (Pinkall & Polthier, 1993).

Given shapes $\mathcal{S}_1$ and $\mathcal{S}_2$, a correspondence $\mathcal{T}: \mathcal{S}_1 \rightarrow \mathcal{S}_2$ is a permutation matrix $\mathbf{P} \in \mathbb{R}^{n_1 \times n_2}$. An initial map $\mathbf{P}^0$ can be refined using pair-wise descriptors $\mathbf{U}_{\mathcal{S}_1}, \mathbf{U}_{\mathcal{S}_2}$:

$$\mathbf{P}^t = \arg \min_{\mathbf{P}^t} \|\mathbf{P}^{t\top} \mathbf{U}_{\mathcal{S}_1} \mathbf{P}^{t-1} - \mathbf{U}_{\mathcal{S}_2}\|, \tag{1}$$

or point-wise descriptors $\mathbf{K}_{\mathcal{S}_1}, \mathbf{K}_{\mathcal{S}_2}$:

$$\mathbf{P}^t = \arg \min_{\mathbf{P}^t} \|\mathbf{P}^{t\top} \mathbf{K}_{\mathcal{S}_1} - \mathbf{K}_{\mathcal{S}_2} f(\mathbf{P}^{t-1})\|. \tag{2}$$

In the functional map framework (Ovsjanikov et al., 2012), $\mathbf{K}$ is a truncated LBO basis $\mathbf{\Phi} \in \mathbb{R}^{n \times k}$, and the functional map $\mathbf{C} \in \mathbb{R}^{k \times k}$ is:

$$\mathbf{C}^t = \arg \min_{\mathbf{C}^t} \|\mathbf{P}^{t\top} \mathbf{\Phi}_{\mathcal{S}_1} - \mathbf{\Phi}_{\mathcal{S}_2} \mathbf{C}^t\|. \tag{3}$$

Imperfect initializations lead to errors in $\mathbf{C}$ (Xiang et al., 2021). The LMD metric selects well-matched points $lks$ to compute:

$$\mathbf{C}^t = \arg \min_{\mathbf{C}^t} \|\mathbf{P}^t(:, lks)^\top \mathbf{\Phi}_{\mathcal{S}_1} - \mathbf{\Phi}_{\mathcal{S}_2}(lks, :) \mathbf{C}^t\|. \tag{4}$$

LMD $\mathbf{D_P} \in \mathbb{R}^{n_1}$ is:

$$[\mathbf{D_P}]_i = \frac{\sum_{j \in \mathcal{B}_\gamma(i)} [\mathbf{M}_{\mathcal{S}_1}]_{i,j} [\mathbf{E}]_{i,j}}{\sum_{j \in \mathcal{B}_\gamma(i)} [\mathbf{M}_{\mathcal{S}_1}]_{i,j}}, \tag{5}$$

where:

$$[\mathbf{E}]_{i,j} = \frac{|[\mathbf{G}_{\mathcal{S}_1}]_{i,j} - [\mathbf{P}^{t\top} \mathbf{G}_{\mathcal{S}_2} \mathbf{P}^t]_{i,j}|}{\gamma}, \tag{6}$$

and $\mathcal{B}_\gamma(i) = \{j \in \mathcal{S}_1 | [\mathbf{G}_{\mathcal{S}_1}]_{i,j} \leq \gamma\}$, and $\mathbf{G}_{\mathcal{S}_1}$ is a matrix of geodesic distances. Prior implementations (Xiang et al., 2021; Kamhoua et al., 2022; Kamhoua & Qu, 2024) used an incorrect $\mathbf{E}$:

$$[\mathbf{E}]_{i,j} = \frac{|[\mathbf{G}_{\mathcal{S}_1}]_{i,j} - [\mathbf{G}_{\mathcal{S}_2} \mathbf{P}^t]_{i,j}|}{\gamma}, \tag{7}$$

assuming pre-aligned rows, leading to errors (Fan et al., 2022). We use the correct form (Eq. 6).

Functional maps may yield inaccurate correspondences due to truncated bases (Kamhoua & Qu, 2024). HOPE (Kamhoua & Qu, 2024) uses seeded graph matching:

$$\mathbf{P}^t(lks, :) = \arg \max_{\mathbf{P}} \mathbf{Tr}(\mathbf{B}_{\mathcal{S}_1, \mathbf{P}^t}(lks, :) \mathbf{P}^{t-1} \mathbf{A}_{\mathcal{S}_2, h}), \tag{8}$$

where $\mathbf{B}_{\mathcal{S}_1, \mathbf{P}^t} = \mathbf{P}^{t\top} \mathbf{A}_{\mathcal{S}_1, h}$, and $\mathbf{A}_{\mathcal{S}_1, h}$ indicates $h$-hop connectivity.

# 4  LOCAL NEIGHBOURHOOD CONSISTENCY

LMD is computationally expensive (near-quadratic due to geodesic distances) and sensitive to non-isometric deformations. We propose the Local Neighborhood Consistency (LNC) metric $\mathbf{N} \in \mathbb{R}^n$:

$$\mathbf{N}_i = \frac{\sum_j |[\mathbf{A}_{\mathcal{S}_1,h}]_{i,j} - [\mathbf{P}^\top \mathbf{A}_{\mathcal{S}_2,h}\mathbf{P}]_{i,j}|}{[\mathbf{D}]_{i,i}}, \tag{9}$$

where $\mathbf{D}$ is the degree matrix. LNC offers:

- **Low Complexity**: Computing $\mathbf{P}^\top \mathbf{A}_{\mathcal{S}_2,h}\mathbf{P}$ involves sparse row/column reordering, yielding $O(n_2 v)$ complexity, where $v$ is the number of nonzero $\mathbf{D}_{i,i}$.
- **Robustness**: LNC relies on connectivity, not geometric properties, making it resilient to non-isometric deformations.
- **Properties**: Due to space, please see Appendix A (will be moved to main paper if accepted).

**Definition 4.1** *An Erdős-Rényi graph $\mathcal{G}(n,p)$ is a random graph on $n$ vertices where each edge is included independently with probability $p$. Let $\mathcal{G}_1, \mathcal{G}_2$ be two graphs derived from an Erdős-Rényi parent graph $\mathcal{G}(n,p)$ with edge correlation $1 - \epsilon$, where $\epsilon \in (0,1)$ represents the noise level. Let $\mathbf{A}_{\mathcal{S}_1,2}$ and $\mathbf{A}_{\mathcal{S}_2,2}$ denote the adjacency matrices of the 2-hop neighborhoods in $\mathcal{G}_1$ and $\mathcal{G}_2$, respectively. Let $\mathbf{P}$ be a permutation matrix in the set of all permutation matrices $\Pi$, and $B$ be a seed set of vertices with alignment fraction $\beta = |B|/n$.*

**Theorem 4.1 (LNC Recovery Guarantee)** *For graphs $\mathcal{G}_1, \mathcal{G}_2$ derived from an Erdős-Rényi parent $\mathcal{G}(n,p)$ with edge correlation $1 - \epsilon$, if the following conditions hold:*

- ***Distinctiveness***: *For any incorrect permutation $\mathbf{P}' \neq \mathbf{P}$,*
  $$\|\mathbf{A}_{\mathcal{S}_1,2} - \mathbf{P}'\mathbf{A}_{\mathcal{S}_2,2}\mathbf{P}'^\top\|_F^2 > \|\mathbf{A}_{\mathcal{S}_1,2} - \mathbf{P}\mathbf{A}_{\mathcal{S}_2,2}\mathbf{P}^\top\|_F^2 + C\epsilon n p^2,$$
  *where $C > 0$ is a constant, and $\|\cdot\|_F$ denotes the Frobenius norm.*

- ***Seed Requirement***: *The seed set size satisfies $\beta|B| = \Omega(\sqrt{n \log n})$.*

*Then, the permutation $\mathbf{P}$ is the unique solution to the optimization problem:*

$$\hat{\mathbf{P}} = \arg\min_{\mathbf{P} \in \Pi} \|\mathbf{A}_{\mathcal{S}_1,2} - \mathbf{P}\mathbf{A}_{\mathcal{S}_2,2}\mathbf{P}^\top\|_F.$$

**Proof 4.1** *To prove that $\mathbf{P}$ is the unique solution, we analyze the optimization problem under the given conditions.*

1. ***Error Bound***: *Consider a vertex pair $(i,j)$ in $\mathcal{G}_1$ and the corresponding pair $(\mathbf{P}(i), \mathbf{P}(j))$ in $\mathcal{G}_2$. The 2-hop degree difference is bounded as:*
   $$|\deg_{2\text{-}hop}(i,j;\mathcal{G}_1) - \deg_{2\text{-}hop}(\mathbf{P}(i),\mathbf{P}(j);\mathcal{G}_2)|$$
   $$\leq \epsilon \deg(i)\deg(j) \leq O(\epsilon n p^2).$$
   *This follows from the edge correlation $1 - \epsilon$, which limits the discrepancy in 2-hop neighborhood structures under connectivity noise.*

2. ***Concentration***: *For edge probability $p \gg n^{-1/2}$, the Frobenius norm of the difference between the aligned 2-hop adjacency matrices concentrates:*
   $$\mathbb{P}(\|\mathbf{A}_{\mathcal{S}_1,2} - \mathbf{P}\mathbf{A}_{\mathcal{S}_2,2}\mathbf{P}^\top\|_F^2 \leq C\epsilon n^2 p^2) \geq 1 - e^{-\Omega(np^2)}.$$
   *This concentration ensures that the error for the correct permutation $\mathbf{P}$ is typically small, with high probability.*

3. ***Recovery***: *Given the seed requirement $\beta|B| = \Omega(\sqrt{n \log n})$, the 2-hop neighborhood statistics provide sufficient information to distinguish the correct permutation. The distinctiveness condition ensures that any incorrect permutation $\mathbf{P}' \neq \mathbf{P}$ incurs a significantly larger error, by at least $C\epsilon n p^2$. Combined with the concentration result, the correct permutation $\mathbf{P}$ minimizes the Frobenius norm with probability at least $1 - n^{-c}$ for some constant $c > 0$, ensuring unique recovery.*

*Thus, $\mathbf{P}$ is the unique solution to the optimization problem.*

## 5 COMPARING LNC TO LMD: RECTANGULAR BOX TO CUBE EXAMPLE

To illustrate the differences between the Local Neighborhood Consistency (LNC) metric and the Local Map Distortion (LMD) metric, consider a demo example where we attempt to match a rectangular box to a cube. Both shapes are represented as 8-vertex meshes, with their vertices forming 8x8 adjacency matrices. The rectangular box has dimensions $2 \times 1 \times 1$, while the cube has uniform side lengths of 1, introducing a non-isometric deformation due to the stretching along one axis. We show why LMD may fail to detect correct correspondences due to its reliance on geodesic distances, while LNC succeeds by leveraging mesh connectivity.

The vertex coordinates for the rectangular box $\mathcal{S}_1$ and the cube $\mathcal{S}_2$ are represented as $8 \times 3$ matrices:

$$
\mathbf{X}_{\mathcal{S}_1} = \begin{bmatrix} 0 & 0 & 0 \\ 2 & 0 & 0 \\ 2 & 1 & 0 \\ 0 & 1 & 0 \\ 0 & 0 & 1 \\ 2 & 0 & 1 \\ 2 & 1 & 1 \\ 0 & 1 & 1 \end{bmatrix}, \quad
\mathbf{X}_{\mathcal{S}_2} = \begin{bmatrix} 0 & 0 & 0 \\ 1 & 0 & 0 \\ 1 & 1 & 0 \\ 0 & 1 & 0 \\ 0 & 0 & 1 \\ 1 & 0 & 1 \\ 1 & 1 & 1 \\ 0 & 1 & 1 \end{bmatrix}.
$$

The adjacency matrices $\mathbf{A}_{\mathcal{S}_1}$ and $\mathbf{A}_{\mathcal{S}_2}$ for both shapes are identical due to their shared topology (a simple cubic mesh with edges connecting adjacent vertices). Given an adjacency matrix $\mathbf{A}_{\mathcal{S}_1,1}$, its 2 hop matrices can be obtained as $\mathbf{A}_{\mathcal{S}_1,2} = \text{bool}\left(\text{bool}\left(\mathbf{A}_{\mathcal{S}_1,1}^2\right) - \mathbf{A}_{\mathcal{S}_1,1} - \mathbf{I}\right)$. We thus have:

$$
\mathbf{A}_{\mathcal{S}_1,1} = \mathbf{A}_{\mathcal{S}_2,1} = \begin{bmatrix} 0 & 1 & 0 & 1 & 1 & 0 & 0 & 0 \\ 1 & 0 & 1 & 0 & 0 & 1 & 0 & 0 \\ 0 & 1 & 0 & 1 & 0 & 0 & 1 & 0 \\ 1 & 0 & 1 & 0 & 0 & 0 & 0 & 1 \\ 1 & 0 & 0 & 0 & 0 & 1 & 0 & 1 \\ 0 & 1 & 0 & 0 & 1 & 0 & 1 & 0 \\ 0 & 0 & 1 & 0 & 0 & 1 & 0 & 1 \\ 0 & 0 & 0 & 1 & 1 & 0 & 1 & 0 \end{bmatrix}, \quad
\mathbf{A}_{\mathcal{S}_1,2} = \mathbf{A}_{\mathcal{S}_2,2} = \begin{bmatrix} 0 & 1 & 1 & 1 & 1 & 1 & 0 & 1 \\ 1 & 0 & 1 & 1 & 1 & 1 & 1 & 0 \\ 1 & 1 & 0 & 1 & 0 & 1 & 1 & 1 \\ 1 & 1 & 1 & 0 & 1 & 0 & 1 & 1 \\ 1 & 1 & 0 & 1 & 0 & 1 & 1 & 1 \\ 1 & 1 & 1 & 0 & 1 & 0 & 1 & 1 \\ 0 & 1 & 1 & 1 & 1 & 1 & 0 & 1 \\ 1 & 0 & 1 & 1 & 1 & 1 & 1 & 0 \end{bmatrix}
$$

where each vertex connects to its three immediate neighbors (e.g., vertex 1 connects to vertices 2, 4, and 5).

Assume a correct permutation matrix $\mathbf{P}$ aligns vertices of $\mathcal{S}_1$ to $\mathcal{S}_2$ (e.g., identity mapping for simplicity, as vertex ordering is consistent). The LNC metric (Eq. 9) computes:

$$
\mathbf{N}_i = \frac{|\sum_j [\mathbf{A}_{\mathcal{S}_1,2}]_{i,j} - [\mathbf{P}^\top \mathbf{A}_{\mathcal{S}_2,2} \mathbf{P}]_{i,j}|}{[\mathbf{D}]_{i,i}} \approx \frac{\sum_j |[\mathbf{A}_{\mathcal{S}_1,1}]_{i,j} - [\mathbf{P}^\top \mathbf{A}_{\mathcal{S}_2,1} \mathbf{P}]_{i,j}|}{[\mathbf{D}]_{i,i}},
$$

where $\mathbf{D}$ is the degree matrix with $\mathbf{D}_{i,i} = 3$ for all vertices (each vertex has three edges). Since $\mathbf{A}_{\mathcal{S}_1} = \mathbf{A}_{\mathcal{S}_2}$ and $\mathbf{P}$ is correct, $\mathbf{P}^\top \mathbf{A}_{\mathcal{S}_2} \mathbf{P} = \mathbf{A}_{\mathcal{S}_1}$, so $\mathbf{N}_i = 0$ for all $i$. This indicates perfect correspondence, as LNC detects that the connectivity structure is preserved, despite the geometric stretching.

In contrast, LMD (Eq. 5) relies on geodesic distances. The geodesic distance matrix $\mathbf{G}_{\mathcal{S}_1}$ for the rectangular box has longer distances along the stretched axis (e.g., between vertices 1 and 2, $\mathbf{G}_{\mathcal{S}_1}(1,2) = 2$), while for the cube, $\mathbf{G}_{\mathcal{S}_2}(1,2) = 1$. For a correct $\mathbf{P}$, the error term (Eq. 6) becomes:

$$
[\mathbf{E}]_{i,j} = \frac{\|[\mathbf{G}_{\mathcal{S}_1}]_{i,j} - [\mathbf{P}^\top \mathbf{G}_{\mathcal{S}_2} \mathbf{P}]_{i,j}\|}{\gamma}.
$$

For vertices along the stretched axis (e.g., $i = 1$, $j = 2$), $[\mathbf{E}]_{1,2} = \frac{|2-1|}{\gamma} = \frac{1}{\gamma}$, yielding a large LMD value, suggesting an incorrect correspondence. LMD fails because the non-isometric deformation alters geodesic distances, even though the topology remains consistent. Thus, LNC correctly identifies the correspondence by focusing on connectivity, while LMD is misled by geometric distortions.

**LNC vs. LMD: Theoretical Comparison.** LMD fails when geodesic distances are altered by non-isometric deformations (e.g., stretching or area changes), as $\mathbf{E}_{i,j}$ becomes large even for correct correspondences. LNC fails when mesh connectivity differs significantly, as $\mathbf{A}_{\mathcal{S}_1,h}$ and $\mathbf{A}_{\mathcal{S}_2,h}$ may not align. For Erdős-Rényi graphs with edge correlation $1 - \epsilon$, LNC's error is bounded by $O(\epsilon n p^2)$, but high $\epsilon$ (e.g., due to remeshing) reduces landmark detection accuracy. LMD is more robust to connectivity changes if geodesics are preserved, but its $O(n^2)$ complexity is prohibitive.

---

**Algorithm 1** NEXUS

---

**Inputs**: LNC threshold $\epsilon$, max iteration $t_{max}$, max hops $h_{max}$, descriptors $\mathbf{K}_{\mathcal{S}_1}$, $\mathbf{K}_{\mathcal{S}_2}$, neighborhood matrices $\mathbf{A}_{\mathcal{S}_1,h}$, $\mathbf{A}_{\mathcal{S}_2,h}$ for $h = [0, \cdots, h_{max}]$
1. Initialize $\mathbf{P}^0$ by solving:

$$\mathbf{P}^0 = \arg\min_{\mathbf{P}^0} \|\mathbf{P}^{0\top}\mathbf{A}_{\mathcal{S}_1,2}\mathbf{K}_{\mathcal{S}_1} - \mathbf{A}_{\mathcal{S}_2,2}\mathbf{K}_{\mathcal{S}_2}\|. \tag{11}$$

**while** $0 \leq t \leq t_{max}$ **do**
    2. Compute LNC to locate landmarks $lks = \{i \in \mathcal{S}_1 | [\mathbf{N}]_i \leq \epsilon(t)\}$, with $h$=2 in Eq.9.
    3. Identify non-landmark points $Nlks$.
    4. Build $\mathbf{A}_{\mathcal{S}_2,h}$ and $\mathbf{A}_{\mathcal{S}_1,h}$ cycling $h$ from 1 to $h_{max}$ following HOPE(Kamhoua & Qu, 2024).
    5. Update $\mathbf{P}^t(Nlks,:)$ using GMWM (YU et al., 2021) to solve Eq. 8.
**end while**
**return** $\mathbf{P}^t$.

---

# 6 NEXUS: NEIGHBORHOOD-ENHANCED CORRESPONDENCE OPTIMIZATION STRATEGY

NEXUS follows a three-step process: (1) initialize correspondences, (2) detect poorly matched points using LNC, and (3) refine matches via seeded graph matching.

**Step 1: Initialization.** We use SHOT descriptors (Tombari et al., 2010) modified by $\mathbf{A}_{\mathcal{S}_1,2}\mathbf{K}_{\mathcal{S}_1}$, $\mathbf{A}_{\mathcal{S}_2,2}\mathbf{K}_{\mathcal{S}_2}$ to enforce connectivity consistency, solving:

$$\mathbf{P}^0 = \arg\min_{\mathbf{P}^0} \|\mathbf{P}^{0\top}\mathbf{A}_{\mathcal{S}_1,2}\mathbf{K}_{\mathcal{S}_1} - \mathbf{A}_{\mathcal{S}_2,2}\mathbf{K}_{\mathcal{S}_2}\|. \tag{10}$$

**Step 2: Detecting Poorly Matched Points.** Landmarks $lks$ are detected as $lks = \{i \in \mathcal{S}_1 | [\mathbf{N}]_i \leq \epsilon(t)\}$, with $\epsilon = \text{linspace}(1.6, 0.6, 10)$ and $\epsilon(t) = 0.6$ for $t \geq 10$.

**Step 3: Refinement.** Non-landmark points $Nlks$ refined (Eq. 8) with GMWM (YU et al., 2021).

**Time Complexity.** Initialization is $O(n \log n)$ via kd-tree (Panigrahy, 2008), LNC computation is $O(n_2 v)$, and GMWM is $O(|Nlks|^2 \log n)$. Total complexity is $O(t|Nlks|^2 \log n)$ for $t$ iterations.

# 7 EXPERIMENTS

We validate NEXUS and LNC on diverse datasets, showing efficiency, effectiveness, and generalization.

## 7.1 EXPERIMENTAL SET-UP

Experiments used Matlab 2023(a) on a Windows 11 system with 32GB RAM and an Intel i5 13500 CPU @ 2.50-4.8GHz.

## 7.2 DATASETS

We evaluate on:

- **TOPKIDS** (Lähner et al., 2017): 25 shapes with up to 12K vertices, featuring near-isometric deformations and topological noise.
- **Re-meshed Datasets**: SCAPE_r, FAUST_r, TOSCA_r, SMAL_r (Cao et al., 2023b), with varied triangulations.
- **Nearly-Isometric Datasets**: SCAPE (Anguelov et al., 2005) (71 shapes, 12.5K vertices) and TOSCA (Bronstein et al., 2008) (80 shapes, 4K–52.5K vertices).
- **Partial Shapes**: SHREC16cuts and SHREC16holes (Bracha et al., 2024; Rodolà et al., 2017), with cuts and holes altering topology.

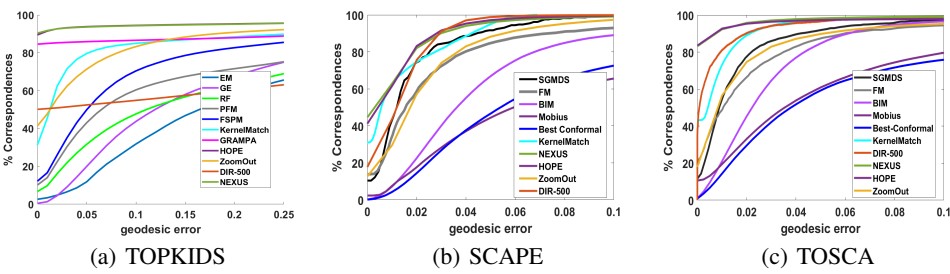

Figure 1: Performance comparison on shapes with topological noise from TOPKIDS 1(a), isometric shapes from SCAPE 1(b) and TOSCA 1(c).

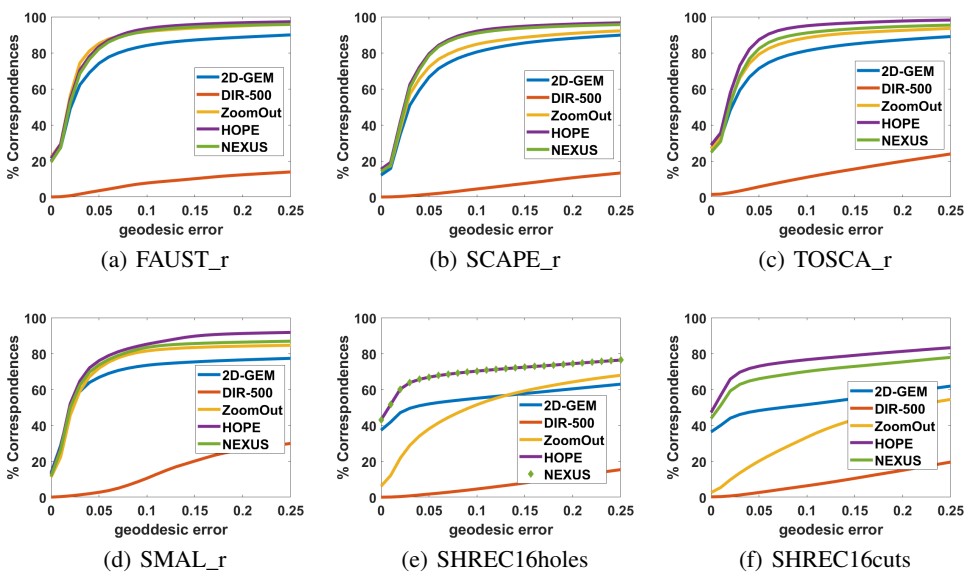

Figure 2: Performance comparison on re-meshed datasets: FAUST_r 2(a), SCAPE_r 2(b), TOSCA_r 2(c), SMAL_r 2(d), and partial datasets: SHREC16cuts 2(f), SHREC16holes 2(e).

### 7.3 EVALUATION METRICS

We use geodesic error (Ehm et al., 2024; Roetzer & Bernard, 2024): $e(i) = \frac{d_{\mathcal{S}_2}(j,j^*)}{\text{diam}(\mathcal{S}_2)}$, where $j$ is the predicted match for $i$, $j^*$ is the ground truth, and $\text{diam}(\mathcal{S}_2)$ is the geodesic diameter.

### 7.4 BASELINES

We compare NEXUS with GRAMPA (Fan et al., 2020), ZoomOut (Melzi et al., 2019), Kernel-Matching (Lähner et al., 2017), HOPE (Kamhoua & Qu, 2024),, SGMDS(Aflalo et al., 2016), FM(Ovsjanikov et al., 2012), BIM(Kim et al., 2011), Mobius(Lipman & Funkhouser, 2009), Best-Conformal (Kim et al., 2011), EM (Sahillioğlu & Yemez, 2012), GE (Lähner et al., 2016), RF (Rodolà et al., 2014), PFM (Rodolà et al., 2017), FSPM (Litany et al., 2017a), DIR (Xiang et al., 2021), and ULRSSM (Cao et al., 2023b).

### 7.5 PARAMETER SETTINGS

For baselines, we follow (Kamhoua & Qu, 2024) using corrected LMD (Eq. 6) where needed. For NEXUS, we use SHOT descriptors (Tombari et al., 2010). Following observations in Sec.A, we set LNC thresholds $\epsilon = \text{linspace}(1.6, 0.6, 10)$, $\epsilon(t) = 0.6$ for $t \geq 10$, $t_{max} = 60$. Following HOPE we set $h_{max} = 8$. For ULRSSM we use their original paper's weights and results.

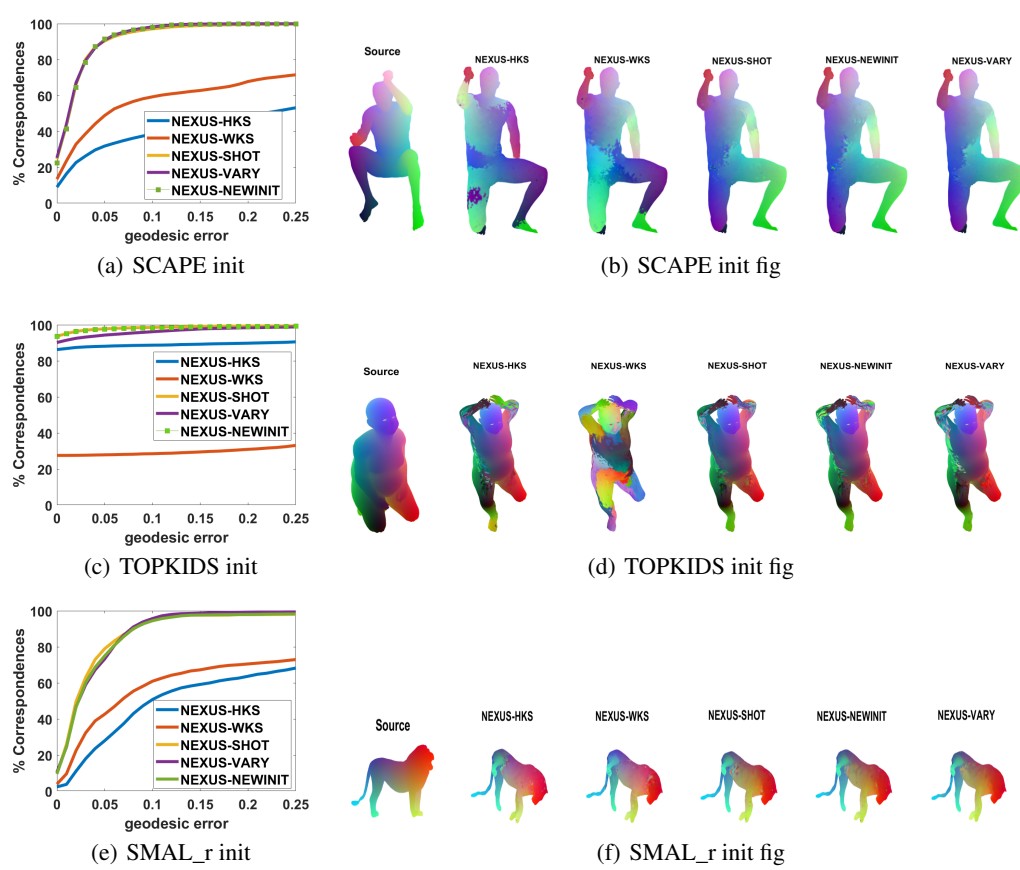

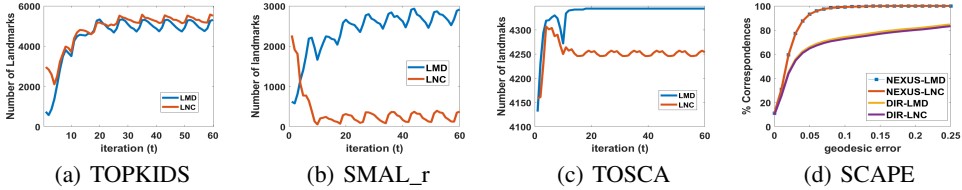

Figure 3: Different initializations on sample shapes from SCAPE, TOPKIDS, and SMAL_r.

Figure 4: Comparison of landmarks detected per iteration by NEXUS using LMD or LNC on TOPKIDS, SMAL_r, and TOSCA. Including performance of LNC vs LMD on SCAPE.

## 7.6 PERFORMANCE ANALYSIS

**Topological Noise.** In TOPKIDS (Fig. 1(a)), NEXUS achieves a precision of 90.41% at geodesic error 0, slightly outperforming the previous best baseline till date HOPE (89.45%) due to the robustness of LNC to non-isometric deformations.

**Nearly-Isometric Shapes.** On SCAPE (Fig. 1(b)) and TOSCA (Fig. 1(c)), NEXUS also outperforms HOPE, and is faster due to LNC's efficiency. For example on matching shape pair 51 on TOSCA (i.e., Michael5 to Michael 7) each shape with around 52.5K vertices, NEXUS takes 655.0 seconds, while HOPE takes 1001.3 seconds. Moreover, the corrected LMD reveals the prior methods' sensitivity to vertex shuffling (Fan et al., 2022) (Fig 1(b)), since the performance degrades.

**Re-meshed Shapes.** On FAUST_r, SCAPE_r, and TOSCA_r (Fig. 2), NEXUS performs comparably to HOPE with corrected LMD, but struggles with significant triangulation differences due to LNC's connectivity reliance.

Table 1: Comparing NEXUS to ULRSSM(Cao et al., 2023b) (a Deep Learning Baseline). The name in brackets attached to ULRSS in the table indicates which dataset it was trained on. Normalized AUC Scores in [0, 1] range reported. Max geodesic error threshold for curve set to 0.2 for all datasets except FAUST_r and SCAPE_r where 0.1 was used.

| Model | SMAL_r | SHRECK16cuts | SHRECK16holes | SCAPE_r | FAUST_r | TOSCA_r | TOPKIDS | SCAPE | TOSCA |
|---|---|---|---|---|---|---|---|---|---|
| NEXUS | .75 | .68 | .61 | .67 | .72 | .83 | .94 | .94 | .98 |
| ULRSSM (FAUST_r+SCAPE_r) | - | - | - | 0.71 | .85 | - | - | - | - |
| ULRSSM (FAUST_r) | - | - | - | 0.78 | .85 | - | - | - | - |
| ULRSSM (SCAPE_r) | - | - | - | 0.81 | .85 | - | - | - | - |
| ULRSSM (SMAL_r) | 0.82 | - | - | - | .85 | - | - | - | - |
| ULRSSM (TOPKIDS) | - | - | - | - | - | - | 0.76 | - | - |
| ULRSSM (SHREC16cuts) | - | .90 | - | - | - | - | - | - | - |
| ULRSSM (SHREC16holes) | - | - | 0.79 | - | - | - | - | - | - |

**Partial Shapes.** On SHREC16cuts and SHREC16holes (Fig. 2), NEXUS and HOPE outperform ZoomOut, as functional maps struggle with altered topologies (Kamhoua & Qu, 2024).

**Time Comparison.** On SCAPE (12.5K vertices), LNC computation takes 0.41s (0.04s for $\mathbf{A}_{\mathcal{S},h}$, 0.37s for Eq. 9), versus 10.42s for LMD (10.05s for $\mathbf{G}_{\mathcal{S}}$, 0.37s for Eq. 5).

**Different Initializations.** Figure 3 demonstrates NEXUS's robustness to various initializations (HKS (Bronstein & Kokkinos, 2010), WKS (Aubry et al., 2011), SHOT (Tombari et al., 2010), and modified SHOT i.e., NEWINIT Eq.11) across SCAPE, TOPKIDS, and SMAL_r datasets, confirming its stability even on challenging datasets. We follow HOPE (Kamhoua & Qu, 2024) for the settings. One can see that the performance remains relatively stable with each different descriptors with the more robust ones like SHOT performing better across the datasets.

**LNC vs LMD.** Figure 4 compares landmarks detected by LMD (Xiang et al., 2021) and LNC (Eq. 9). LNC consistently detects more landmarks as accuracy improves, validating its effectiveness. On SMAL_r (Fig. 4(b)), LNC struggles with inconsistent triangulations, indicating potential for threshold adjustments. On TOPKIDS, where LMD struggles due to topological noise, LNC outdoes LMD by detecting more landmarks faster and helping the algorithm outperform HOPE (Fig 1(a)). Moreover, Fig. 4(d) shows LNC matches or outperforms LMD while being much faster (as discussed in time comparison above).

**Deep Learning.** Table 1 shows that NEXUS significantly outperforms the baseline when meshes have consistent triangulations even in the presence of topological noise (e.g., TOPKIDS). Moreover, it can be observed that though NEXUS struggles with re-meshed and partial shapes compared to the deep learning baseline, it nonetheless generalizes better since the same algorithmic pipeline works across unlike the deep learning baseline that needs retraining and inference time adaptation.

**Ablation and Sensitivity.** As shown in Appendix Sec. B (Figure 5), NEXUS — despite using an untuned $\epsilon$ schedule — outperforms all variants across isometric, non-isometric, and re-meshed shapes. Ablations confirm LNC's necessity for shape matching (NEXUS-3/5 fail without it), while both LNC and k-hop refinement degrade under severe re-meshing due to neighborhood inconsistency (Kamhoua & Qu, 2024).

**Limitations of Nexus.** NEXUS's reliance on mesh connectivity fails when triangulations are highly inconsistent (YU et al., 2021). Its quadratic complexity $O(n^2 \log n)$ in the worst case is higher than linear methods (Melzi et al., 2019; Xiang et al., 2021).

## 8 CONCLUSION

We introduced NEXUS, featuring the LNC metric for efficient and robust shape correspondence. NEXUS outperforms baselines on diverse datasets, offering a fast, generalizable solution. Future work could address connectivity sensitivity and reduce complexity.

## 9 REPRODUCIBILITY AND IMPACT STATEMENT

The code implementing NEXUS is attached in the supplementary material, together with the corrected code for the baselines that used LMD. This code, together with Alg. 1 and the Experiments (Sec. 7) can help reproduce the paper's method and contribution. NEXUS enhances shape correspondence for graphics applications, simplifying and accelerating pipelines without negative impacts.

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

# NEXUS: Neighborhood-Enhanced Correspondence Optimization Strategy for Shape Correspondences Appendices:

## A  PROPERTIES OF THE LOCAL NEIGHBORHOOD CONSISTENCY METRIC

The Local Neighborhood Consistency (LNC) metric for vertex $i$ is defined as:

$$\mathbf{N}_i = \frac{\sum_j \left| [\mathbf{A}_{\mathcal{S}_1,h}]_{i,j} - [\mathbf{P}^\top \mathbf{A}_{\mathcal{S}_2,h} \mathbf{P}]_{i,j} \right|}{[\mathbf{D}]_{i,i}}, \tag{12}$$

where $\mathbf{A}_{\mathcal{S}_1,h}$ and $\mathbf{A}_{\mathcal{S}_2,h}$ are binary $h$-hop adjacency matrices, $\mathbf{P}$ is the correspondence matrix, and $[\mathbf{D}]_{i,i} = \sum_j [\mathbf{A}_{\mathcal{S}_1,h}]_{i,j}$ denotes the degree of vertex $i$ in the $h$-hop graph of $\mathcal{S}_1$.

**Maximum Value**  Since both adjacency matrices are binary, each term in the numerator is either 0 or 1. The summation is bounded above by the total number of neighbors of vertex $i$ in $\mathcal{S}_1$, which is exactly $[\mathbf{D}]_{i,i}$. Thus,

$$\mathbf{N}_i \leq \frac{[\mathbf{D}]_{i,i}}{[\mathbf{D}]_{i,i}} = 1. \tag{13}$$

Therefore, **the maximum value $\mathbf{N}_i$ can attain is 1**.

**When Maximum Occurs**  The value $\mathbf{N}_i = 1$ occurs if and only if, for every vertex $j$ that is within $h$ hops of $i$ in $\mathcal{S}_1$, the corresponding vertex $\mathbf{P}(j)$ in $\mathcal{S}_2$ is *not* within $h$ hops of $\mathbf{P}(i)$ — and vice versa where applicable. More precisely, the binary neighborhood indicators are completely anti-correlated over the local support:

$$\forall j, \quad [\mathbf{A}_{\mathcal{S}_1,h}]_{i,j} \neq [\mathbf{P}^\top \mathbf{A}_{\mathcal{S}_2,h} \mathbf{P}]_{i,j} \quad \text{whenever} \quad [\mathbf{A}_{\mathcal{S}_1,h}]_{i,j} = 1. \tag{14}$$

This may arise due to:

- Grossly incorrect local correspondences in $\mathbf{P}$,
- Structural features (e.g., handles, holes, boundaries) present in one shape but absent in the other.

**Interpretation and Significance**  A value of $\mathbf{N}_i = 1$ signifies **complete local inconsistency** in the $h$-hop neighborhood structure under the current correspondence $\mathbf{P}$. This is a strong indicator that:

- The match at vertex $i$ is unreliable or erroneous,
- The region around $i$ may require correction, exclusion from refinement, or special handling (e.g., in seeded matching or outlier rejection),
- There may be a fundamental structural mismatch between the two shapes at this location.

Conversely, $\mathbf{N}_i = 0$ indicates perfect local consistency — an ideal correspondence preserving neighborhood topology.

**Practical Implications**  Due to its bounded range $[0, 1]$, computational efficiency, and topological robustness, LNC serves as an effective *confidence score* for point-wise correspondences. High LNC values ($\approx 1$) can be used to:

- Filter out unreliable matches in refinement pipelines (cf. "lks" selection in Eq. 4),
- Guide sampling in learning-based frameworks,
- Detect regions of non-isometry or topological discrepancy between shapes.

Thus, LNC not only quantifies local correspondence quality but also enables adaptive, structure-aware shape matching even under challenging deformations.

# B  ABLATION AND PARAMETER SENSITIVITY STUDIES

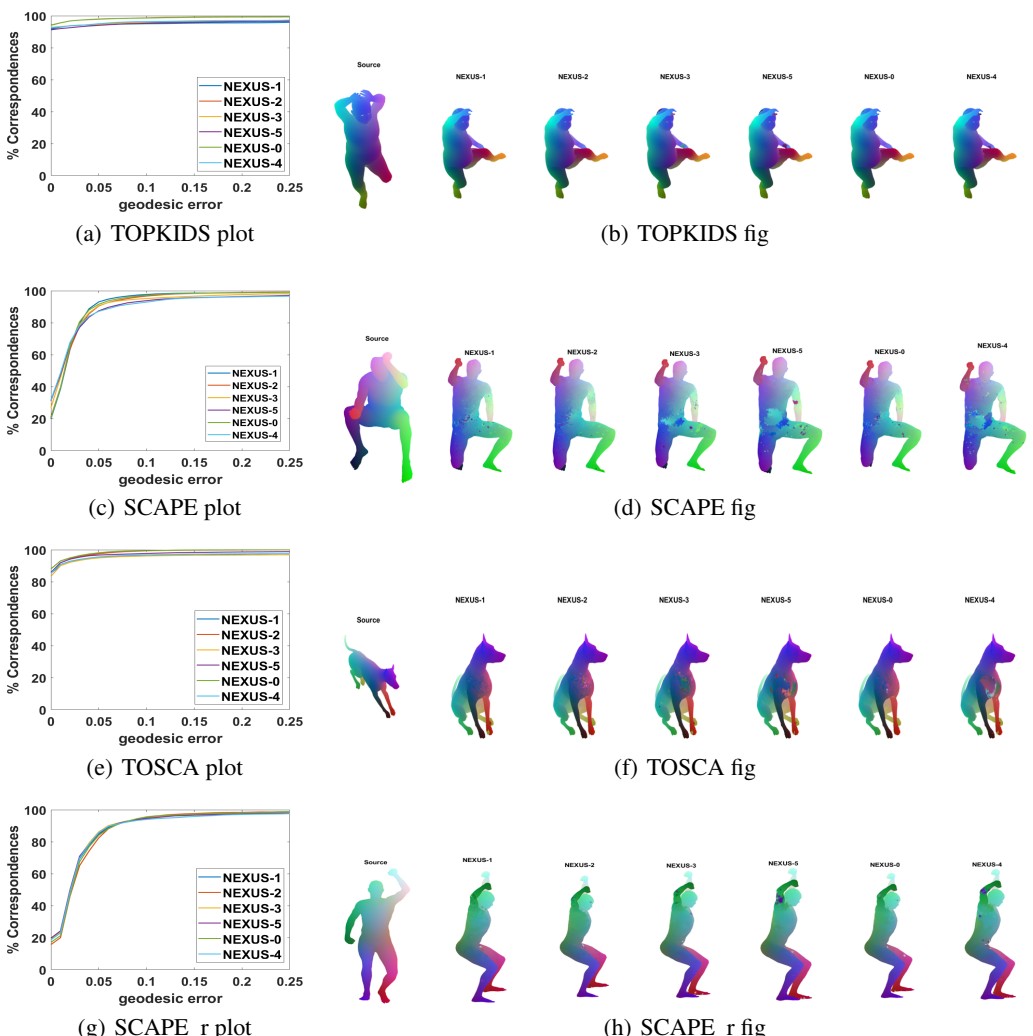

Figure 5: Ablation and sensitivity studies on TOPKIDS 5(a), SCAPE 5(c), TOSCA 5(e), and SCAPE_r 5(g).

In this section, we conduct parameter and ablation studies. We follow HOPE (Kamhoua & Qu, 2024) and use the following settings:

- NEXUS-0: Algorithm 1 with $t_{max} = 60$ and $\epsilon = linespace(100, 0.2, 10)$,

- NEXUS-1: where we reduce the number of iterations to $t_{max} = 20$ in Alg. 1,

- NEXUS-2: where we set $\epsilon = linespace(1, 0.2, 10)$ in Alg. 1,

- NEXUS-3: where we simply solve Eq. 8 with $h = 1$ and $h = 2$ alternatively per iteration in Alg. 1,

- NEXUS-4: where use $h_{max} = 2$ in Alg. 1,

- NEXUS-5: where we simply solve Eq. 8 with $h = [1, 2, \cdots, 8]$ alternatively per iteration in Alg. 1.

From Figure 5, NEXUS-0 emerges as the overall best-performing variant, achieving top results on both isometric (Figure 5(c)) and non-isometric (Figure 5(a)) shapes, while matching other variants on re-meshed shapes (Figure 5(g)). NEXUS-2 ranks second. This suggests that the $\epsilon$ range used in our main experiments (Section 7) — $linspace(1.6, 0.2, 10)$, chosen based on $\max(\mathbf{N}_i)$ (Section A)

without parameter tuning — is sub-optimal; starting with a higher $\epsilon$ appears beneficial when initial landmark estimates are inaccurate.

Moreover, variants without LNC (NEXUS-5 and NEXUS-3) perform poorly on isometric shapes (Figure 5(c)), underscoring LNC's critical role in mutually refining landmark and non-landmark correspondences, especially in handling symmetries. On re-meshed shapes, while performance remains reasonable, both LNC and k-hop neighborhood refinement (Section 6) show limited robustness to severe neighborhood inconsistencies in mesh pairs (Kamhoua & Qu, 2024).

