# OpenReview forum: "NEXUS: Neighborhood-Enhanced Correspondence Optimization Strategy for Shape Correspondences"
_ICLR.cc/2026/Conference — ICLR 2026 Conference Withdrawn Submission_

### Official Review · Reviewer_8Ssj · 2025-10-23

**Soundness:** 3
**Presentation:** 2
**Contribution:** 2
**Rating:** 2
**Confidence:** 4

**Summary:**

The paper proposes an optimization-based technique for solving non-rigid shape matching on meshes. The paper introduces the Local Neighborhood Consistency (LNC) quantity, which, by my intuition, measures the h-hop neighborhood distortion induced by the mapping. Then, it is integrated into NEXUS, a method that starts from some descriptors (SHOT), computes regions with high LNC error, and then uses a graph matching technique (GMWM) to refine the results. NEXUS is tested on meshes (SCAPE, FAUST, TOPKIDS, SMAL, SHREC16 partial shapes), showing results close to the state of the art.

**Strengths:**

- Neighborhood preservation is a reasonable property to enforce
- The paper presents a self-contained definition of the problem and preliminaries

**Weaknesses:**

- The contribution seems quite limited. The paper's main novelty lies in a metric for neighborhood preservation in terms of connectivity, which is then integrated into previous approaches. However, neighborhood preservation is a well-known property to enforce. The paper does not discuss the line of work in combinatorial matching, such as [1,2,3,4], where neighborhood preservation and connectivity play an important role. Its improvement is not particularly notable compared to the baselines. Finally, since the method highly relies on topology, it suffers from remeshing. From Figure 2, the method seems to perform worse than HOPE, the baseline that inspires NEXUS.

- Comparison provided by the paper in terms of curves makes it difficult to analyze the actual values, which could be useful to check that previous methods have been replicated consistently with previous works. I also believe that comparison with recent combinatorial methods listed above would be meaningful and in line with the approach proposed by the paper. It would also be interesting to compare on more recent datasets, like DT4D [5] (largely used in the shape matching community) and BeCoS [6] for partial shapes.

- The paper presentation does not meet ICLR standards. The method discussion does not provide intuitions, and it is overcrowded with heavy notation. Related works are quite packed, and look like a list of methods without providing a positioning of the current work. Figures are not well-curated; for example, the curves in Figure 1 are crowded by methods that are particularly old (e.g., BIM), and every plot contains a legend in a different order. The text contains typos (e.g., line 301, missing square; Tab. 1 "SHRECK" instead of "SHREC" and "ULRSS" instead of "ULRSSM", also some entries are reported as "0.61" and others just ".61"), and there are duplicated references (the paper "Unsupervised learning of robust spectral shape
matching" appears twice, and also "Diffusionnet: Discretization agnostic learning on surfaces"). Section 5 discusses a toy example to give an intuition of the proposed approach. While I appreciate the intent, I found it more suited for supplementary material, and maybe decorated with a visualization figure. I believe all this makes the paper difficult to access for readers.


[1]: Roetzer, P., & Bernard, F. (2025). Fast Globally Optimal and Geometrically Consistent 3D Shape Matching. arXiv preprint arXiv:2504.06385.

[2]: Roetzer, P., Abbas, A., Cao, D., Bernard, F., & Swoboda, P. (2024, September). Discomatch: Fast discrete optimisation for geometrically consistent 3d shape matching. In European Conference on Computer Vision (pp. 443-460). Cham: Springer Nature Switzerland.

[3]: Roetzer, P., Ehm, V., Cremers, D., Lähner, Z., & Bernard, F. (2025). Higher-Order Ratio Cycles for Fast and Globally Optimal Shape Matching. In Proceedings of the Computer Vision and Pattern Recognition Conference (pp. 21793-21803).

[4]: Ehm, V., Roetzer, P., Eisenberger, M., Gao, M., Bernard, F., & Cremers, D. (2024, March). Geometrically consistent partial shape matching. In 2024 International Conference on 3D Vision (3DV) (pp. 914-922). IEEE Computer Society.

[5]: Li, Y., Takehara, H., Taketomi, T., Zheng, B., & Nießner, M. (2021). 4dcomplete: Non-rigid motion estimation beyond the observable surface. In Proceedings of the IEEE/CVF International Conference on Computer Vision (pp. 12706-12716).

[6]: Ehm, V., Amrani, N. E., Xie, Y., Bastian, L., Gao, M., Wang, W., ... & Bernard, F. (2025, August). Beyond Complete Shapes: A Benchmark for Quantitative Evaluation of 3D Shape Surface Matching Algorithms. In Computer Graphics Forum (Vol. 44, No. 5, p. e70186).

**Questions:**

1) How does the paper position w.r.t. [1,2,3,4] from above? How does it compare to them, both in terms of performance and matching quality?
2) Do you think it is possible to extend the method to the partial-to-partial case?
3) Could you provide results on DT4D or BeCoS shapes?
4) Remeshing seems to be a critical issue for the method; could you provide further analysis on how much remeshing impacts the results? (e.g., from uniform meshes to highly non-isotropic remeshing)

---

### Official Review · Reviewer_QWj1 · 2025-10-28

**Soundness:** 2
**Presentation:** 1
**Contribution:** 2
**Rating:** 2
**Confidence:** 4

**Summary:**

This paper presents an iterative correspondence refinement algorithm, replacing geodesic-based aggregation scores (LMD) with an h-hop connectivity score (LNC). At each iteration, landmarks are selected via LNC and the full map is refined using an existing seeded graph matching solver on.

Claimed benefits are lower runtime (no dense geodesics) and robustness to non-isometric deformations (no geometric information preservation).

The method is evaluated on several standard shape datasets with ablations. The paper also notes and fixes an error in a prior LMD implementation.

This is a conceptually simple swap from geodesic distance to adjacency inside a graph matching pipeline. However, the paper’s framing and exposition makes it hard to assess the true novelty and scope.

**Strengths:**

LNC is faster to compute than LMD as it discards all geometric information, making it a cheap and useful replacement for LMD.

Using seeded graph matching is an reasonable for shape matching, and perhaps underlooked.

The experiments span multiple datasets and include ablations. Limitations under remeshing are acknowledged.

**Weaknesses:**

**Overall Readability and Exposition**

I found the paper a bit hard to follow on a first read. Section 2 uses a lot of acronyms, and key objects are not introduced before being used. In particular, Section 3 introduces several variables  ($f(P_{t-1})$, $lks$, $D_P$) without clear definitions or context.  Important properties of LNC are deferred to the appendix ("Properties: Due to space, please see Appendix A").

Section 5’s toy example relies on showing fulll adjacency matrices without a clear visual. It also only works in the identical-triangulation setting and doesn’t reflect nor discuss the sensitivity to remeshing here. I think there is a deeper question to ask about using graph matching for geometric shapes (see below).

**Framing and Scope**

In essence the approach is a seeded graph matching algorithm: computing per-vertex score from connectivity (LNC) to pick seeds, then refine the map by seeded GM. The paper should state this plainly and position itself within the seeded-GM literature, rather than presenting it primarily as a new geometric-matching principle.

- The paper suggests that **discarding geometry is better for geometric matching** ("LNC relies on connectivity, not geometric properties, making it resilient to non-isometric deformations", Section 4). There is no consensus that throwing away geometric information improves shape matching. I'd even say that generally, high sensitivity to the triangulation is seen as a major issue in the literature. As written, the claim is way too strong and not backed by the experimental results.
- Methodologically, the framework appears to mirror existing graph matching baselines (e.g. HOPE), with the main change being to swap the geodesic distances with h-hop adjacency matrices. That’s a fine idea, but it should be stated explicitly, with a clear positioning with similar baselines.

**Results**

Most figures are correspondence curves with many overlapping lines, which makes it hard to read off improvements. The claim of outperforming the closest baseline (HOPE) is difficult to verify from the plots alone.

Include quantitative numbers, and qualitative comparisons for multiple methods, not just the proposed one.

There is a mention of sensitivity to triangulation, but no discussion or results regarding matching fixed geometry with varying triangulations.


**Technical Precision**

- The paper treats the matrices $P$ as permutation matrices (Section 3, Section 4) which is not the case in practice with varying number of vertices. It is unclear what kind of matrix $P$ is used in practice.
- Section 4 mixes dimension: $P^\top A_2 P$ works if $P$ is $n_2 \times n_1$, but $P$ is the opposite size as per Section 3
- Eq (5) is internally not consistent: The paper defines $M$ as a diagonal mass matrix earlier, then uses $M_{ij}$  in a neighbor average in Eq(5) . If $M$ is diagonal, off-diagonals vanish, and the formula collapses. Please clarify whether you intended $M_{ii}$, and fix the equation accordingly.
- It is unclear what Def 4.1 actually defines. It seems to be a mix between notations and definitions
- Theorem 4.1 is not introduced nor discussed. It is unclear what its assumptions require, and how results on ER graph can relate or translate to shape meshes. I am also suspecting that there might be a typo: since $\beta = \frac{|B|}{n}$ already, I think the "seed requirement" part in Thm 4.1 concerne $|B|$ rather than $\beta |B|$
- "Modified SHOT" is mentioned but never defined
- Original dataset papers for all datasets used **are not cited** (only one work that used them).
- There are multiple typos that add friction.

**Questions:**

1 - In what precise ways does your pipeline differ from HOPE beyond swapping LMD for LNC and the "modified SHOT" init?

2 - Are you optimizing over a binary assignment, a doubly-stochastic relaxation, or a true permutation ?

3 - What is "Modified SHOT" ?

4 - Please restate Theorem 4.1 with precise assumptions and consequences. Also please provide a short explanation of how the ER-graph model informs shape graphs. I believe there is very little to share between the two.

5 - Concretely, what should a practitioner take away from Theorem 4.1?

6 - Do you believe that a connectivity-only graph-matching approach is the right general strategy for shape correspondence? If so, under what assumptions (sampling, triangulation noise, ...) do you expect it to outperform methods that use metric/descriptor information?

---

### Official Review · Reviewer_iEQq · 2025-10-29

**Soundness:** 2
**Presentation:** 1
**Contribution:** 2
**Rating:** 2
**Confidence:** 5

**Summary:**

The paper proposes a new metric to measure the correspondence quality, namely Local Neighborhood Consistency (LNC), which is more computationally efficient and robust compared to previous metric, e.g. Local Map Distortion (LMD). Moreover, the paper integrates LNC with a seeded graph matching approach to build an iterative optimization shape matching framework. i.e. NEXUS framework. The paper demonstrates that the proposed NEXUS framework simplifies and accelerates shape matching pipeline while maintaining or improving accuracy compared to baseline methods.

**Strengths:**

1. The proposed metric LNC is simple and efficient, it replaces the geodesic distance matrix, which is computationally expensive, by adjacency matrix and degree matrix in LMD.
2. It conducts theoretical analysis of LNC to prove that the solution is unique under certain assumptions and builds an analytical comparison between LNC and LMD, emphasizing that LNC is more robust for non-isometric shape matching compared to LMD.

**Weaknesses:**

1. The paper is not well-written. In the introduction section, the paper directly introduces the primary contributions while lacks the motivation part. The related work section also solely mention a bunch of previous works without mentioning their strengths and limitations and the connection to the proposed method. In the experiment section, the paper only shows the PCK curves without quantitative evaluation, such as AUC (area under curve) and the baseline comparison is also not consistent (see Figure 1 and Figure 2).
2. In Figure 3, the paper demonstrates the NEXUS performance under different initializations, while it is unclear what is NEXUS-VARY. From the results in Figure 3, it seems the method is still sensitive to different initializations. The performance degrades a lot with WKS and HKS initialization.
3. The method utilizes adjacency matrix as the measurement of the correspondence quality, which makes the method very sensitive to the mesh connectivity, so the method is likely to fail for meshes with different triangulation, which is quite common in real-world settings.

**Questions:**

1. What is the landmark detection experiment shown in Figure 4? Could you explain the experiment setting in details?
2. Table 1 compares the proposed method with an unsupervised learning method, while it still demonstrates that the proposed method has worse performance compared to learning-based method. So, what is the strength of the proposed method compared to learning-based method?

---

### Official Review · Reviewer_fz7o · 2025-10-31

**Soundness:** 2
**Presentation:** 2
**Contribution:** 2
**Rating:** 2
**Confidence:** 4

**Summary:**

This paper proposes a framework for nonrigid shape matching. Their framework is based on their proposed local neighborhood consistency (LNC) metric that measures mesh quality using the 2-hop adjacency matrix of the mesh. This metric is computationally more efficient than the prior Local Map Distortion (LMD) metric that relies on mesh area and geodesic distances. Their algorithm optimizes the map in the presence of descriptors (SHOT in their case) iteratively using seeded graph matching. They experimentally validate their method on several datasets and conduct ablations over the input features.

**Strengths:**

This paper builds on a new direction in shape matching, using N-hop neighborhoods and treating the input meshes as graphs. To this end, they demonstrate robustness to different types of shapes including meshes with topological noise, nearly-isometric shapes, re-meshed shapes, and partial shapes. Their work is based on Theorem 4.1, which proves that the computed permutation matrix is the unique solution to the optimization problem that matches the 2-hop neighborhood of the two meshes. The theoretical guarantee is a strength of this paper. Further, their optimization algorithm is simple in practice and appears to be somewhat faster than other classical approaches.

The paper is somewhat original, though it builds on the work of HOPE (Kamhoua and Qu, NeurIPS 2024) that proposed aligning shapes by aligning k-hope neighborhoods. The LNC metric is a novel contribution that overcomes the limitations of LMD. However, LMD is not by any means the standard distortion metric used in shape correspondence. They also repaired the previously proposed LMD metric, which is a contribution for future works based on this.

**Weaknesses:**

### Claims
There are certain claims made in the paper that I disagree with and seem unfounded:
- the authors give the impression that the LMD metric is widely used and is at the core of many shape correspondence works. However, this is simply not true. Why make the entire paper focused on addressing this metric? There is a large class of shape correspondence algorithms that do not use it.
- The authors emphasize the computational efficiency of LNC, claiming that it is linear in complexity. However, the metric is O(nv), where n is the number of nonzero entries in the adjacency matrix (basically the number of edges), and v is the number of vertices. How is this not a quadratic complexity?
- The authors claim that deep learning-based mapping methods would need to be retrained. However, many DL-based works generalize quite well to a diverse class of shapes. There are even works that propose zero-shot matching (by test-time optimization, see [1], for example).
- The claim that deep learning methods do not do well on partial shapes is also not true.
- The improvement in computational wall clock time seems overblown. There is only 2 time numbers reported for pairs of meshes, and that is not enough to make a conclusion. Further, the paper does not even mention the speed of the deep learning-based shape matching methods. There are other classical methods, such as [2], that are somewhat fast and provide strong maps.

### Limited Evaluation
- The authors only report the geodesic error in their experiments. Additional metrics, for example, conformal distortion, looking at overlap in segmentation (when available)
- Why are there so many blank entries on Table 1?
- There are more works on deep learning-based matching that is not include. For example, [3].
- Can you also include figures of the lower quality meshes and partial shapes?
- Overall, the evaluation is far too limited. The authors only picked a handful of meshes and claim that this method is superior to other mapping approaches.

### Methodological concerns
- Why is Equation (9) the right optimization problem to solve? As far as I see, this metric does not take any information about the mesh into account, and converts the mesh into a graph.


### Clarity of Writing
- Can the authors provide intuition behind equations (5) and (6)? What exactly are these trying to measure? What would minimize this metric?
- Is there supposed to be a term Pt in Equation 3? P is not an optimization variable, and P is typically derived from C.

Overall, I feel that there are too many dubious claims, that the author are not solving a major problem in shape matching, and that evaluation is limited.

## References
[1] Attaiki, Souhaib, and Maks Ovsjanikov. "NCP: Neural correspondence prior for effective unsupervised shape matching." Advances in Neural Information Processing Systems 35 (2022): 28842-28857.

[2] Ezuz, Danielle, Justin Solomon, and Mirela Ben-Chen. "Reversible harmonic maps between discrete surfaces." ACM Transactions on Graphics (ToG) 38.2 (2019): 1-12.

[3]Cao, Dongliang, et al. "Spectral meets spatial: Harmonising 3d shape matching and interpolation." Proceedings of the IEEE/CVF Conference on Computer Vision and Pattern Recognition. 2024.

**Questions:**

- How would this work extend to soft mapping problems (i.e. where you compute a vertex-to-face or vertex-to-tet map), rather than a permutation matrix? For example see [1,2] for these types of maps.
- I find it surprising that the SHOT features work the best, I would have expected WKS or HKS. Can the authors give some intuition why?


### References
[1]Ezuz, Danielle, Justin Solomon, and Mirela Ben-Chen. "Reversible harmonic maps between discrete surfaces." ACM Transactions on Graphics (ToG) 38.2 (2019): 1-12.

[2]Abulnaga, S. Mazdak, et al. "Symmetric volume maps: Order-invariant volumetric mesh correspondence with free boundary." ACM Transactions on Graphics 42.3 (2023): 1-20.

---

### Note · Authors · 2025-11-19

**Comment:**

# Withdrawal of ICLR 2026 Submission #7165 – NEXUS

Dear Area Chair,

I am writing to formally withdraw submission #7165 titled “NEXUS: Neighborhood-Enhanced Correspondence Optimization Strategy for Shape Correspondences” from ICLR 2026.

We are grateful to the four reviewers and the area chair for the significant time and effort invested in evaluating our work. The reviews primarily highlight two recurring points: (1) the need for broader and more quantitative experimental validation (additional metrics, more baselines, more datasets, and clearer figures), and (2) requests for improved clarity in theoretical aspects, notation, and positioning relative to seeded graph-matching and combinatorial methods.

While we do not share all of the concerns—particularly regarding the scope and strength of the experimental evaluation—we will carefully review each comment to determine how each suggestions can further strengthen the manuscript and revise it accordingly in the future.

Thank you once again for the detailed feedback.

God bless,
Barakeel Fanseu Kamhoua
(on behalf of all co-authors)

**Withdrawal Confirmation:**

I have read and agree with the venue's withdrawal policy on behalf of myself and my co-authors.